# Genome-Wide Identification of Polyamine Oxidase (PAO) Family Genes: Roles of *CaPAO2* and *CaPAO4* in the Cold Tolerance of Pepper (*Capsicum annuum* L.)

**DOI:** 10.3390/ijms23179999

**Published:** 2022-09-02

**Authors:** Jianwei Zhang, Le Liang, Jiachang Xiao, Yongdong Xie, Li Zhu, Xinru Xue, Linyu Xu, Peihan Zhou, Jianzhao Ran, Zhi Huang, Guochao Sun, Yunsong Lai, Bo Sun, Yi Tang, Huanxiu Li

**Affiliations:** 1College of Horticulture, Sichuan Agricultural University, Chengdu 611130, China; 2Institute for Processing and Storage of Agricultural Products, Chengdu Academy of Agricultural and Forest Sciences, Chengdu 611130, China; 3Institute of Pomology and Olericulture, Sichuan Agricultural University, Chengdu 611130, China

**Keywords:** pepper, PAO gene family, cold stress, overexpression, prokaryotic expression

## Abstract

Polyamine oxidases (PAOs), which are flavin adenine dinucleotide-dependent enzymes, catalyze polyamine (PA) catabolism, producing hydrogen peroxide (H_2_O_2_). Several PAO family members have been identified in plants, but their expression in pepper plants remains unclear. Here, six PAO genes were identified in the ‘Zunla-1’ pepper genome (named *CaPAO1*–*CaPAO6* according to their chromosomal positions). The PAO proteins were divided into four subfamilies according to phylogenetics: CaPAO1 belongs to subfamily I; CaPAO3 and CaPAO5 belong to subfamily III; and CaPAO2, CaPAO4, and CaPAO6 belong to subfamily IV (none belong to subfamily II). *CaPAO2*, *CaPAO4*, and *CaPAO6* were ubiquitously and highly expressed in all tissues, *CaPAO1* was mainly expressed in flowers, whereas *CaPAO3* and *CaPAO5* were expressed at very low levels in all tissues. RNA-seq analysis revealed that *CaPAO2* and *CaPAO4* were notably upregulated by cold stress. CaPAO2 and CaPAO4 were localized in the peroxisome, and spermine was the preferred substrate for PA catabolism. *CaPAO2* and *CaPAO4* overexpression in *Arabidopsis thaliana* significantly enhanced freezing-stress tolerance by increasing antioxidant enzyme activity and decreasing malondialdehyde, H_2_O_2_, and superoxide accumulation, accompanied by the upregulation of cold-responsive genes (*AtCOR15A*, *AtRD29A*, *AtCOR47*, and *AtKIN1*). Thus, we identified candidate PAO genes for breeding cold-stress-tolerant transgenic pepper cultivars.

## 1. Introduction

Cold stress is an adverse abiotic factor that not only influences the geographical distribution of plants, but also negatively impacts crop growth, development, productivity, and quality. Cold stress can be divided into chilling (0–15 °C) and freezing (<0 °C) stresses, according to the impact on plants under different temperature intensities [1]. Generally, plants of different origins exhibit different sensitivities to cold stress. For example, *Arabidopsis* and other plants originating from temperate regions can exhibit improved cold resistance through prior exposure to low (non-freezing) temperatures—a process known as cold acclimation [1,2]. However, most tropical and subtropical crops are sensitive to cold stress. To minimize the damage caused by stress, plants possess several adaptive mechanisms to regulate their gene expression and metabolic levels, which improve the likelihood of survival under adverse conditions.

Polyamines (PAs), including putrescine (Put), spermidine (Spd), and spermine (Spm), are small aliphatic amines with low molecular weights that are commonly found in both prokaryotic and eukaryotic cells. PAs are closely related to growth, development, and biotic and abiotic stress responses in higher plants [3,4,5,6]. Cellular PA levels maintain a dynamic balance through biosynthetic and catabolic pathways, involving two types of oxidases—diamine oxidase (DAO or CuAO) and polyamine oxidase (PAO) [7,8]; DAO catalyzes Put and Spd degradation, producing ammonia and hydrogen peroxide (H_2_O_2_) and PAO can contribute to terminal catabolism (TC) and back-conversion (BC), according to their reaction modes. In the TC process, PAOs convert Spm or Spd to N-(3-aminopropyl)-4-aminobutanal or 4-aminobutanal, respectively, producing H_2_O_2_ and 1,3-diaminopropane (DAP) [7,9,10]. The initially characterized PAOs in monocotyledonous plants exhibit functions in the TC pathway; examples include maize PAO (ZmPAO) [11], barley PAO (HvPAO1 and HvPAO2) [12], and rice PAO (OsPAO2, OsPAO6, and OsPAO7) [13,14]. Citrus CsPAO4 is also involved in the TC pathway [15]. Another reported BC process is the opposite of the PA synthesis pathway, which converts Spm to Spd and/or Spd to Put and usually produces H_2_O_2_ and 3-aminopropanal [16]. All PAOs in *Arabidopsis* (AtPAO1–AtPAO5) participate in the BC pathway [17,18,19]. Several PAOs in rice (OsPAO1 and OsPAO3-OsPAO5) [20,21], upland cotton (GhPAO3) [22], and tomato (SlPAO2–SlPAO5) have been reported to contribute to this pathway [23].

The signaling molecule H_2_O_2_ is a common product of both TC and BC pathways. PAOs regulate plant growth and development through H_2_O_2_. For example, Tisi et al. [24] found that the overexpression of the *ZmPAO* gene in tobacco promoted root differentiation and induced programmed cell death. Wu et al. [25] reported that AtPAO oxidizes Spd to produce H_2_O_2_ in *Arabidopsis*, which signals Ca^2+^ influx and regulates pollen tube growth. Furthermore, Spd-derived apoplastic H_2_O_2_ induces stress-responsive gene expression and programmed cell death [26]. Moreover, PAO-derived H_2_O_2_ is associated with biotic and abiotic stress responses. Gémes et al. [27] reported that the overexpression of the maize *ZmPAO* gene in tomato improves the ability of plants to resist salt stress. In *Arabidopsis*, *AtPAO1*, *AtPAO2*, and *AtPAO3* are responsive to salt stress, and double mutant (*AtPAO1* and *AtPAO5*) plants have improved salt and drought stress resilience [28]. In contrast, *CsPAO4* negatively regulates the salt tolerance of sweet orange, and *CsPAO4* overexpression in tobacco results in diminished salt resistance [15]. Furthermore, *AtPAO1* and *AtPAO2* expression was found to be upregulated in *Arabidopsis* infected with *Pseudomonas syringae*, and the double mutant *AtPAO1/2* plants were more susceptible to disease [29]. Despite these findings, the role of PAO in cold stress response remains unclear.

Pepper (*Capsicum annuum* L.) belongs to the Solanaceae family and is an important vegetable and raw material for spices, medicine, and cosmetics worldwide. Pepper seedlings are specifically vulnerable to cold stress in early spring. Previously, a multiomics analysis demonstrated that the expression levels of two genes encoding polyamine oxidase (LOC107859655 and LOC107878076) in pepper increased significantly under cold stress, indicating that these genes may regulate cold stress response [30]. Here, we identified six CaPAO genes in pepper and analyzed the evolutionary tree, gene structure, and cis-elements in the promoter. Furthermore, two CaPAO genes, *CaPAO2* and *CaPAO4*, were isolated, and their biological functions in the cold stress response were analyzed. These results provide new insights into the cold-stress response mechanisms in pepper plants and form a basis for improving cold tolerance in new pepper varieties.

## 2. Results

### 2.1. Identification of CaPAO Family Genes in Pepper

After removing redundant and erroneous sequences, six PAO genes were identified in the ‘Zunla-1’ genome and named *CaPAO1*–*CaPAO6* according to their chromosomal positions. Among these, *CaPAO1, CaPAO2, CaPAO3*, and *CaPAO4* are located on different chromosomes (Chr.) (Chr.1, Chr.2, Chr.5, and Chr.7 respectively), while *CaPAO5* and *CaPAO6* are located on Chr.9 (Appendix A). The full-length CaPAO sequences varied from 489 to 657 amino acids (aa), with the protein weight ranging from 54.5 to 72.5 kilodalton (KDa), respectively. The isoelectric point (pI) of the protein ranged from 4.74 to 5.91. The instability index values were less than 40, indicating that all pepper CaPAO proteins were stable. The in silico subcellular localization of CaPAO proteins was predicted using the CELLO website. CaPAO2, CaPAO4, and CaPAO6 were predicted to be located in peroxisomes, whereas CaPAO1, CaPAO3, and CaPAO5 were identified outside the cell regions (Table 1).

### 2.2. Sequence Alignment and Signal Peptide Analysis

Multiple sequence alignment was performed between the six CaPAO sequences of pepper and five sequences of *Arabidopsis*. The results showed a significant difference among these PAO protein sequences (Figure 1). AtPAO2 and AtPAO3 were the most identical (84.73% identify), while AtPAO1 exhibited the lowest sequence similarity, of 21.81%, to CaPAO3. Among the six CaPAOs, the homology between CaPAO4 and CaPAO6 was the highest at 83.71%, followed by that between CaPAO2 and CaPAO4 at 65.33% and between CaPAO2 and CaPAO6 at 64.67%. The homology of other Capsicum CaPAO sequences was less than 50%, among which the sequence homology between CaPAO2 and CaPAO3 was only 22.81%, indicating that these two sequences displayed a low degree of conservation.

The peroxisomal targeting signal (PTS), which acts as a tripeptide at the C-terminus of proteins, can be efficiently imported into peroxisomes. In plants, the preferred PTS1 tripeptides are SKL>, SRL>, AKL>, SRM>, ARL>, and SKM> (“>” indicates the C-terminal end of the peptide) [31]. Consistent with online software predictions, CaPAO2, CaPAO4, and CaPAO6 were predicted to be localized in plant peroxisomes containing a PTS1 tripeptide (SRM). All pepper PAOs possessed a trans-membrane domain, while only CaPAO2, CaPAO4, and CaPAO6 contained a signal peptide (Figure 1).

### 2.3. Phylogenetics, Motif, Gene Structure, and Cis-Element Analysis

To understand the evolutionary relationships among CaPAO genes, a phylogenetic tree was constructed based on the PAO protein sequences of different species, including *Arabidopsis*, tomato, *Brassica rapa*, *Brachypodium distachyon*, rice, maize, and pepper. As shown in Figure 2A, all PAO proteins clustered into four subfamilies (I, II, III, and IV). The CaPAO sequences of pepper were distributed in subfamilies I, III, and IV, similar to AtPAO members in *Arabidopsis*. Among them, CaPAO1 belonged to subfamily I, CaPAO3 and CaPAO5 were classified into subfamily III, and CaPAO2, CaPAO4, and CaPAO6 were categorized into subfamily IV. Subfamily II had no CaPAO members in pepper plants. Notably, CaPAOs were clustered together with tomato, *Arabidopsis*, and *B. rapa* PAOs, suggesting that pepper PAO proteins were more closely related to those from tomato, *Arabidopsis*, and *B. rapa* than to those of maize, rice, and *B. distachyon*.

A total of 11 motifs were identified from CaPAO protein sequences in pepper using the MEME online software. According to the frequency of occurrence, motifs 2, 3, 4, 6, 8, and 9 were the most frequently presented in all CaPAO proteins. Motifs 1, 10, 5, and 7 were observed in subfamily IV, whereas motif 11 only existed in subfamily III (Figure 2B). In addition, gene structure analysis showed that the members of subfamily I and subfamily IV all contained 10 exons and 9 introns, whereas members of subfamily III only contained one exon and did not have any introns (Figure 2C). These results indicate that CaPAOs are evolutionarily complex and may have different functions.

To better understand the potential function of the CaPAO gene, a 2000 bp sequence upstream of the promoter was analyzed by the Plant CARE website. A total of 48 cis-elements, related to stress responses and phytohormones, were identified (Appendix A). Phytohormone-related elements mainly included abscisic acid, gibberellin, methyl jasmonate, auxin, and salicylic acid, which were widespread in different CaPAO genes. Related to abiotic stress responsivity, drought-inducibility elements were distributed in the *CaPAO2* and *CaPAO3* genes, while the elements of low-temperature response were located in the sequences of *CaPAO2*, *CaPAO4*, and *CaPAO5* (Figure 2D).

### 2.4. Expression of CaPAOs in Different Tissues and in Response to Cold Stress

To better understand the role of CaPAOs in pepper development, we examined their transcript levels using quantitative real-time polymerase chain reaction analysis (qRT-PCR). Samples were collected from three different stages, including seedling (root:R-Dev1, stem:S-Dev1, leaf:L-Dev1, and cotyledon), flowering (flower), and fruiting (immature fruit, F-Dev2; mature fruit, F-Dev3; root, R-Dev3; stem, S-Dev3; and leaf, L-Dev3). We found that CaPAO genes in the same subfamily exhibited similar tissue-specific expressions (Figure 3A). Three genes, *CaPAO2*, *CaPAO4*, and *CaPAO6*, all of which belong to subfamily IV, were highly expressed in S-Dev1, L-Dev1, flower, F-Dev3, and L-Dev3 but exhibited lower expression levels in F-Dev2, R-Dev3, and S-Dev3. The expression levels of subfamily III (*CaPAO3* and *CaPAO5*) members were lower than those of subfamily IV in all tissues, except for the expression level of the *CaPAO3* gene in S-Dev1 and S-Dev3. Notably, *CaPAO5* displayed very low transcript levels in all tissues, especially in L-Dev1. The residual *CaPAO1*, belonging to subfamily I, showed distinct expression patterns, and its expression level was highest in flowers, followed by S-Dev1 and R-Dev1.

In addition, we analyzed the transcription level of CaPAO genes in response to cold stress using previous transcriptome data [30]. Compared with the control, *CaPAO2* and *CaPAO4* gene expression levels first increased and then decreased throughout the treatment period, but they were significantly higher than control levels, except for the expression level of *CaPAO4* at 24 h. There were no significant differences between the remaining genes and the control group (Figure 3B). Considering these findings in combination with the previous results for cis-elements, we focused on the *CaPAO2* and *CaPAO4* genes in response to cold stress.

### 2.5. Prokaryotic Expression of CaPAO2 and CaPAO4 and Substrate Specificity of Their Enzymes

According to previous experiments, the isopropyl-β-d-thiogalactoside (IPTG) concentrations that induced the expression of CaPAO2 and CaPAO4 recombinant proteins were 0.6 and 0.4 mM, respectively (data not shown). Compared with the empty (lane 1) and non-induced (lane 2) control, the protein bands of CaPAO2 and CaPAO4, with molecular weights of 54.8 and 54.7 kDa, respectively, were detected in lane 3. After ultrasonic fragmentation, the supernatant (lane 4) contained bands of the target protein, indicating that CaPAO2 and CaPAO4 were soluble proteins. The purified CaPAO2 and CaPAO4 showed a single band (lane 6) with molecular weights of approximately 54.8 and 54.7 kDa, respectively (Figure 4A,B), and the concentrations were measured as 0.22 and 0.33 mg/mL, respectively. The spectral absorption of CaPAO2 and CaPAO4 in the range of 300–540 nm revealed that there were two absorption maxima at approximately 380 and 460 nm, respectively, indicating a typical feature of FAD-contained proteins (Figure 4C,D). The enzymatic activities of recombinant CaPAO2 and CaPAO4 were determined using Put, Spd, and Spm. CaPAO2 and CaPAO4 oxidized Spd and Spm, but the preferred substrate was Spm. Put was found to be a poor substrate (Figure 4E,F).

### 2.6. Analysis of CaPAO2 and CaPAO4 Subcellular Localization

To further elucidate the subcellular localization of CaPAO2 and CaPAO4, pAN58-GFP-CaPAO2/4 fusion vectors were constructed (Figure 5A). The laser confocal microscopy results showed that pAN58-GFP-CaPAO2 and pAN58-GFP-CaPAO4 fusion proteins were located in peroxisomes and overlapped with the red marker signal, whereas the green fluorescent protein (GFP) signal of the positive control filled the entire protoplast (Figure 5B). In addition, we constructed pAN58-CaPAO2/4-GFP fusion vectors driven by the 35S promoter; however, no fluorescence signal was detected using the same method (results not shown).

### 2.7. CaPAO2 and CaPAO4 Overexpression Improved Freezing Tolerance of Transgenic Arabidopsis

To better understand the function of *CaPAO2* and *CaPAO4* genes in cold tolerance, the floral-dip method was used to obtain transgenic *Arabidopsis*. Homozygous T3 lines were obtained using kanamycin resistance selection and genomic PCR verification. Three lines with the highest expression levels of *CaPAO2* (OE-4, OE-8, and OE-8) and *CaPAO4* (OE-1, OE-7, and OE-8) were identified for subsequent cold-tolerance experiments (Appendix A).

Throughout the entire growth cycle of *Arabidopsis*, no significant differences between transgenic lines and wild-type (WT) plants were observed (3 weeks). After 5 weeks of growth, the plant height and number of lateral branches of the transgenic lines were lower than those of the WT plants. After 6 weeks, the inflorescences of the CaPAO4-OE lines were significantly lower than those of the WT, but there was no significant difference between the CaPAO2-OE lines and WT (Appendix A). These phenotypes indicate that *CaPAO2* and *CaPAO4* affect the growth and development of *Arabidopsis* plants.

We further used *Arabidopsis* as a model to carry out chilling (4 °C) and freezing (−8 °C) treatments. Figure 6A shows that no differences were observed between the WT and transgenic plants under control conditions and chilling stress. After freezing stress, the leaves of most WT plants wilted, and the color of the leaves deepened, showing an obvious frostbite phenotype, whereas the leaves of some transgenic plants were normal. After 3 days of recovery at room temperature, almost all WT plants died, whereas the CaPAO2/4-OE lines were still alive. The survival rates of the CaPAO2-OE (68–75%) and CaPAO4-OE (48–65%) lines were significantly higher than WT plants (20%) (Figure 6B,C).

### 2.8. CaPAO2 and CaPAO4 Overexpression Affects PAs under Cold Stress

PAs play important roles in the regulation of plant cold responses [4]. Under room temperatures, there were significant differences in the content of all PAs between CaPAO2/4-OE lines and WT plants, except for the Put content of CaPAO4-OE lines. After chilling stress (4 °C), the content of Spm in CaPAO2-OE lines and Spd in CaPAO2/4-OE lines were significantly higher than those in WT plants, whereas the Put content in CaPAO2-OE lines was distinctly lower than that in WT plants. Compared with WT plants, only the Spd content of CaPAO2/4-OE lines was significantly higher after freezing stress (−8 °C), and the CaPAO2-OE lines were 7.2–7.46 times that of WT plants (Figure 7A–C), indicating that the sharp increase in Spd content in CaPAO2/4-OE lines plays an important role in response to cold stress.

### 2.9. Overexpression of CaPAO2 and CaPAO4 Alters MDA and ROS Content and Antioxidant Enzyme Activity during Cold Stress

Physiological indices, such as malondialdehyde (MDA), H_2_O_2_, and O_2_^−^ contents, are commonly used to assess the stress tolerance of plants in response to abiotic stressors. Under room temperature and 4 °C chilling treatment, no significant differences in the above physiological indices were observed between CaPAO2/4-OE lines and WT plants, except for the H_2_O_2_ content of CaPAO2-OE lines under chilling stress. After freezing stress (−8 °C), the MDA, H_2_O_2_, and O_2_^−^ in WT plants were significantly higher than those in the CaPAO2/4-OE lines (Figure 8A–C). As an antioxidant enzyme that catalyzes the conversion of O_2_^−^ to H_2_O_2_ and O_2_, CAT is widely distributed in plant tissues and catalyzes H_2_O_2_ decomposition into H_2_O and O_2_. Compared with WT plants, SOD and CAT activities in the CaPAO2-OE lines were significantly increased after chilling stress, but there was no significant difference in the CaPAO4-OE lines. After freezing stress, the SOD, POD, and CAT activities of CaPAO2/4-OE lines were significantly higher than those of WT plants (Figure 8D–F). These results demonstrate that CaPAO2/4 overexpression enhanced resistance to freezing by increasing the activities of antioxidant enzymes and decreasing the MDA, H_2_O_2_, and O_2_^−^ accumulation in transgenic plants under freezing stress conditions.

### 2.10. CaPAO2 and CaPAO4 Overexpression Activated the Expression of Cold-Responsive Genes under Cold Stress

We analyzed the expression levels of cold-responsive (COR) genes in CaPAO2/4-OE lines and WT plants. As shown in Figure 9A–D, the expression levels of these four genes in all plants were relatively low at room temperature. After cold stress, the expression levels of COR genes increased in both the CaPAO2/4-OE lines and WT plants. However, there were significant differences between the expression of these genes in the transgenic lines and WT plants after freezing stress. These results showed that *CaPAO2* and *CaPAO4* could activate the expression of *AtCOR15A*, *AtRD29A*, *AtCOR47*, and *AtKIN1*, enhancing the freezing resistance of transgenic *A. thaliana* lines.

## 3. Discussion

The dynamic balance of PAs, which is primarily dependent on anabolism and catabolism, plays an important role in the growth, development, and stress responses of higher plants [3,4,5,6]. PAO participates in catabolic pathways, such as the TC and BC pathways. The genes encoding PAO proteins constitute a small gene family [33]. To date, several members of the PAO gene family have been identified in many plants (e.g., 7, 7, 7, 7, 6, 6, 6, 5, 5, and 4 PAO genes in tomato, grape, tea, rice, apple, sweet orange, *B. rapa*, *B. distachyon*, *Arabidopsis*, and peach, respectively) [17,23,32,34,35,36,37,38,39]. However, little is known about the PAO family in pepper. In this study, six PAO genes distributed across five chromosomes were identified in pepper. The molecular weight, protein length, and protein weight of these genes were similar; however, significant differences in the protein sequence were detected, which is consistent with other plant species [17,23,32,38]. Three pepper proteins (CaPAO2, CaPAO4, and CaPAO6) contain a PTS consisting of three amino acid residues (SRM sequence) at the C-terminal end of the protein, which has been reported to target proteins to peroxisomes in plants [40]. In this study, we connected CaPAO2 and CaPAO4 to the N-terminus of GFP for subcellular localization, which confirmed that CaPAO2 and CaPAO4 were located in peroxisomes (Figure 5). However, the following conclusion cannot be reached when the gene is connected to the C-terminus of the fluorescent labeling, which is consistent with previous studies by Moschou et al. [18]. This may be because the peroxisome signals of CaPAO2 and CaPAO4 are affected by GFP; therefore, the peroxisome fluorescence signal could not be detected.

A phylogenetic tree was constructed using pepper, *Arabidopsis*, tomato, rice, and other plants, which were clustered into four subfamilies, consistent with previous reports [17,23,35,36,38]. The members of the CaPAOs were divided into subfamilies I, III, and IV, without subfamilies II (Figure 2A). Proteins of subfamilies I, III, and IV, such as *Arabidopsis* AtPAO1–AtPAO5, rice OsPAO1, OsPAO3, and OsPAO5, and tomato SiPAO2–SiPAO5, can catalyze the BC-reaction of PAs. Among them, most members of subfamily III are located in peroxisomes, which can scavenge H_2_O_2_ through the abundant peroxisomal catalase, also providing effective evidence for their involvement in the BC pathway [18,19]. The last subfamily II members are monocotyledonous plants that have the ability to catalyze TC reactions, such as rice OsPAO2, OsPAO6, OsPAO7, and maize ZmPAO. Therefore, we hypothesize that all CaPAOs oxidize PAs in the BC pathway, especially CaPAO2 and CaPAO4, which are located in peroxisomes. However, further studies are required to confirm this.

To identify the specific expression levels of CaPAO genes in pepper, qRT-PCR was used to determine gene expression in roots, stems, leaves, flowers, and fruits at different growth stages. Compared with subfamilies I and III, the members of subfamily IV, including CaPAO2, CaPAO4, and CaPAO6, were ubiquitously and highly expressed in all tissues (Figure 3A), which is consistent with the results obtained for tomatoes, rice, and tea plants [20,23,35]. This result shows that these three genes may play an important role in the growth and development of pepper. In addition, we showed that drought-inducibility elements existed in the promoters of the *CaPAO2* and *CaPAO3* genes, whereas three genes (*CaPAO2*, *CaPAO4*, and *CaPAO5*) contained low-temperature response elements (Figure 2D). RNA-seq data analysis showed that the expression of the *CaPAO2* and *CaPAO4* genes was upregulated after cold stress (Figure 3B). Based on the above results, we conclude that *CaPAO2* and *CaPAO4* participated in the regulation of cold stress in pepper seedlings.

We successfully generated the bacterial strain BL21 (DE3)-pCOLD-CaPAO2/4, which expressed CaPAO2 and CaPAO4. BL21 (DE3)-pCOLD-CaPAO2/4 cells were induced with hypothermia, and the cells were lysed to obtain CaPAO2 and CaPAO4 proteins. These proteins were verified by 12% SDS-PAGE, in which protein bands corresponding to CaPAO2 and CaPAO4 proteins were observed (Figure 4A,B). To further understand the characteristics of these two proteins, we analyzed their preferences for PA substrates. Ono et al. found that Spm was the best substrate for OsPAO4 and OsPAO5 in rice and Spd was the optimal substrate for OsPAO3, but none of these proteins used Put as a substrate [20]. In *B. distachyon*, analyzing BdPAO2 revealed that the preferred substrate was Spd, while BdPAO3 preferred Spm [39]. In addition, Spd and Spm significantly induced CsPAO4 activity in sweet oranges, which was not altered by Put [15]. In our study, both Spd and Spm were oxidized by CaPAO2 and CaPAO4 proteins, all of which preferred Spm. Consistent with previous results, Put could not be a substrate of CaPAO2 and CaPAO4 proteins (Figure 4E,F). These results indicate that CaPAO2 and CaPAO4 can oxidize and decompose Spd and Spm in plants and that the dynamic balance of PAs is primarily maintained through changes in Spm.

After transformation of *A. thaliana* with *CaPAO2* and *CaPAO4*, the WT and transgenic lines were subjected to chilling (4 °C) and freezing (−8 °C) stress. After chilling stress, all plants were undamaged. However, after freezing stress, the degree of damage in WT plants was more severe than that in CaPAO2/4-OE lines (Figure 6A). The survival rate of CaPAO2/4-OE lines was higher than that of the WT plants (Figure 6B,C). The results showed that *CaPAO2* and *CaPAO4* improved freezing stress response in transgenic pepper plants to a certain extent.

To explore the mechanism by which these two genes increase cold tolerance in plants, it is necessary to determine their physiological, biochemical, and molecular levels. The cold resistance of plants is related to changes in PA content. For example, Huang et al. [41] reported that the cold resistance of transgenic tobacco is higher than that of WT, which is accompanied by a significant increase in PA content. Compared with WT plants, the Spd content of the CaPAO2/4-OE lines increased significantly, but there was no significant difference in Spm content, which is consistent with the results of Zhuo et al. [42] MDA, an end product of membrane lipid peroxidation, acts as an indicator of the degree of damage to plant tissue. Reactive oxygen species (ROS), such as H_2_O_2_ and O_2_^−^, have a two-way effect: one acts as part of a cellular signaling network to regulate plant physiology and metabolism [25], and the other is a toxic by-product of aerobic metabolism, which affects plant growth [24,26]. In the present study, the MDA, H_2_O_2_ and O_2_^−^ in CaPAO2/4-OE lines were significantly lower than those in WT plants under freezing stress, indicating that the plants were less damaged by stress (Figure 8A–C). Interestingly, CaPAO2 overexpression in *Arabidopsis* lines resulted in higher contents of H_2_O_2_ under chilling stress, which may be related to the decomposition of PAs catalyzed by CaPAO2. Protective enzymes, such as SOD, POD, and CAT, can remove excessive ROS to maintain intracellular balance [43]. Antioxidant enzyme activity is positively correlated with cold tolerance [44,45,46]. COR genes, such as *AtCOR15A*, *AtRD29A*, *AtCOR47*, and *AtKIN1*, contain one or more conserved cis-elements, which are regulated by the C-repeat binding factor. In trifoliate orange, ICE1 combines with the arginine decarboxylase (ADC) gene to improve plant cold resistance by increasing PAs content, antioxidant enzyme activity, and COR gene expression level [41]. The results showed that compared with WT plants, the activity levels of SOD, POD, and CAT in CaPAO2/4-OE line significantly increased under freezing stress. This is consistent with higher COR gene expression level. These results indicate that transgenic *Arabidopsis* with *CaPAO2* and *CaPAO4* has a stronger ability to scavenge ROS and to promote the expression of *AtCOR15A*, *AtRD29A*, *AtCOR47*, and *AtKIN1*, which improves resistance to freezing. However, it remains unknown how *CaPAO2* and *CaPAO4* participate in the cold-resistant regulatory network to improve COR gene expression, and further research and confirmation are needed.

## 4. Materials and Methods

### 4.1. Plant Material and Treatments

Gan Zi pepper plants were used in this study. Seeds were selected and grown in a 50-hole plug tray containing a mixture of peat: vermiculite: perlite (2:1:1, *v*/*v*/*v*) in a greenhouse under 25 °C light (16 h)/20 °C dark (8 h) cycles until six true leaves emerged. The seedlings were separated into two groups; plants in the first group were subjected to part collection for cotyledon, root, stem, and leaf samples, and plants in the second group were transplanted into a 26 × 26 cm flowerpot filled with substrate (nutrient soil: vermiculite = 3:1) and then transferred to a greenhouse for the remainder of the growth cycle. The growth period was from June to September 2020. Samples of flowers, immature fruits, mature fruits, roots, stems, and leaves of fruit maturity were collected. All samples were immediately refrigerated in liquid nitrogen and stored at −80 °C until further use.

Columbia-0 (Col-0, *Arabidopsis thaliana*) was used for genetic transformation in this study. WT and Transgenic seeds were soaked and placed in a refrigerator at 4 °C for vernalization for 3 days, then sown in a 9 × 9 cm nutrient bowl filled with high-pressure sterilized seedling substrate, coated with film, and germinated. The culture conditions were as follows: 25 °C light (16 h) and 20 °C dark (8 h). To study the cold stress tolerance of transgenic lines, T3 generation *Arabidopsis* and WT plants for 3–4 weeks were used for the experiment. Plants were subjected to chilling stress (4 °C) for 24 h and freezing stress (−8 °C) for 6 h. After freezing stress, treated plants were moved to normal conditions for 3 days. The plant survival rate was determined after recovery.

### 4.2. Identification of CaPAO Genes in Pepper

The protein sequences of AtPAO (*Arabidopsis*) and OsPAO (*Oryza sativa*) were downloaded from the NCBI database and used as queries to perform BLASTP search with a cutoff value of 1 × 10^−10^. PFAM (https://pfam.xfam.org/search#tabview=tab1; accessed on 6 October 2021) and CD-search (https://www.ncbi.nlm.nih.gov/Structure/bwrpsb/bwrpsb.cgi; accessed on 6 October 2021) websites were used to remove redundant sequences with Amino_oxidase domain (PF01593).

### 4.3. Bioinformatics Analysis of CaPAO in Pepper

The physicochemical properties of the CaPAO amino acids were analyzed using the ExPASy website (http://web.Expasy.org/protparam/; accessed on 1 December 2021). Putative locations were detected using the CELLO website (http://cello.life.nctu.edu.tw/; accessed on 1 December 2021). The domain sequences in pepper and maize were aligned by ClustalX software (1.83) [47]. The parameters were a gap opening penalty of 10.00; a gap extension penalty of 0.20; delay divergent sequences of 30%; a DNA transition weight of 0.5; and no use of a negative matrix. The full-length proteins of *Arabidopsis*, tomato, pepper, rice, maize, *B. rapa*, and *B. distachyon* were aligned using ClustalW [47], and a phylogenetic tree was constructed using MEGA-X software together with the maximum likelihood method (Bootstrap method with 1000 replications was selected and default values used for other parameters). Motifs in CaPAO proteins were identified using the MEME Suite website (http://meme-suite.org/; accessed on 25 April 2022), with the maximum number of motifs set at 11 and default values used for other parameters. The gene structure was analyzed by the GSDS website (http://gsds.cbi.pku.edu.cn/; accessed on 17 December 2021). The sequences of 2000 bp 5′ upstream in the CaPAO genes were obtained using the TBtools software [48]. These sequences were further analyzed using the Plant Care website (http://bioinformatics.psb.ugent.be/webtools/plantcare/html/; accessed on 7 December 2021).

### 4.4. Gene Expression Analysis

Total RNA was extracted using the RNAprep Pure Plant Plus Kit (TIANGEN, Beijing, China), and cDNA synthesis was performed using the PrimeScript™ RT reagent Kit (TaKaRa, Dalian, China), according to the manufacturer’s instructions. qRT-PCR was performed using 2 × SYBR Green Fast qPCR Mix (Biomarker, China) on a CFX96 real-time PCR system (Bio-Rad, Hercules, CA, USA). The primers used in this study were designed using Primer3 (http://primer3.ut.ee/ (accessed on 17 November 2020)). The relative expression levels of CaPAO genes and *Arabidopsis* genes were calculated using the 2^−ΔCt^ and 2^−ΔΔCt^ methods, respectively [49,50]. CaUbi3 and Atactin2 were used as the reference gene for pepper and *Arabidopsis*, respectively [43,51]. All primer sequences are shown in Appendix A. Preliminary RNA-seq data [30] were used to determine the expression of the CaPAO genes in response to cold stress.

### 4.5. CaPAO Gene Isolation

According to the experimental requirements, CaPAO2 and CaPAO4 were amplified via PCR from cDNA of young pepper seedlings using gene specific primers (Appendix A). The amplified PCR products were cloned into the pEASY-Blunt Simple Cloning Vector (TransGen Biotech, Beijing, China), and the sequences were confirmed.

### 4.6. Preparation of Recombinant CaPAO Protein in Escherichia coli

The above cloning vectors were digested with EcoRI and SaI1 enzymes (TransGen Biotech, Beijing, China) and then cloned into the prokaryotic expression vector pCold I (TaKaRa, Tokyo, Japan), resulting in pCold-CaPAOs. After sequencing, pCold-CaPAOs were transformed into *E. coli* BL21 (DE3) cells (TaKaRa, Dalian, China). The recombinant proteins were expressed following the pCold vector manufacturer’s recommendations (Takara, Dalian, China), and the concentration of IPTG was 0.4 or 0.6 mM. Cells were collected by centrifugation at 10,000× *g* for 10 min, resuspended in 10 mM phosphate buffer (PBS, pH 7.0), and disrupted by sonication on ice. After centrifugation at 12,000× *g* for 20 min at 4 °C, the cleared supernatant was mixed with Ni-NTA agarose (TransGen Biotech, Beijing, China). The resin was washed with wash buffer (50 mM NaH_2_PO_4_, 300 mM NaCl, 10 mM imidazole, 10 mM Tris base, pH 8), and the recombinant protein was eluted using an elution buffer (50 mM NaH_2_PO_4_, 300 mM NaCl, 10 mM imidazole, 10 mM Tris base, pH 8). The fusion protein was detected by 12% SDS-PAGE. The protein concentration was detected using the Easy Protein Quantitative Kit (TransGen Biotech, Beijing, China).

### 4.7. CaPAO Activity Assay

The activities of recombinant CaPAO2 and CaPAO4 proteins were determined according to a previously reported method [17,52,53]. In 1 mL of the reaction system, approximately 5 μg protein was added to 100 mM phosphate buffer (pH 7.0), 100 μM 4-aminoantipyrine, 1 mM 3,5-dichloro-2-hydroxybenzesulfonic acid, 0.06 mg/mL horseradish peroxidase, and 5 μL 10 mM substrate (Put, Spd, and Spm) to initiate the reaction. The activities of the proteins were measured for 2 min at 515 nm using a UV-1200 spectrophotometer (Mapada, Shanghai, China).

### 4.8. Subcellular Localization Analysis

The open reading frame (ORF) sequences of *CaPAO2* and *CaPAO4* were separately cloned into the pAN58 vector using the XbaI and PstI restriction endonucleases. The GFP was linked to the C-terminus of the target gene to form GFP-CaPAO2/4. pAN58-GFP was used as a control, and the hydroxypyruvate reductase gene of *Arabidopsis* was used as a peroxisome marker [54]. The extraction and transformation of protoplasts were performed according to the method outlined by Yoo et al. [55]. Fluorescence signals were observed using a Nikon C2-ER confocal laser scanning microscope (Nikon Instruments, Tokyo, Japan).

### 4.9. Arabidopsis Transformation

To generate the pBWA(V)HS-CaPAO2/4 overexpression vector, the ORF regions of the CaPAO2/4 genes were cloned into a BsmBI/Esp3I-digested pBWA(V)HS vector using golden gate technology. These two fusion constructs were transformed into WT *Arabidopsis* Col-0 using the floral-dip method, and the seeds were harvested [56]. Homozygous transgenic lines were screened on Murashige and Skoog (MS) medium containing 50 mg/L kanamycin. The T3 generation plants were identified by PCR, and gene expression was analyzed by qRT-PCR. All primer sequences are listed in Appendix A.

### 4.10. Determination of Physiological Indicators and Polyamine Content in Arabidopsis

Free PA (Put, Spd, Spm) content was determined as previously described [30]. MDA content was determined using a plant MDA assay kit (Nanjing Jiancheng, Nanjing, China). H_2_O_2_ and superoxide (O_2_^−^) contents were determined using a H_2_O_2_ content assay kit and superoxide anion activity assay kit (Solarbio, Beijing, China), respectively, according to the manufacturer’s instructions.

SOD, POD, and CAT activities were determined following the procedure described by Zhou et al. [57] with slight modifications. Briefly, 0.3 g leaf samples were homogenized on ice in 1.6 mL of 50 mM PBS (pH 7.8) and then centrifuged at 10,000× *g* for 25 min at 4 °C. The supernatant (hereafter referred to as the enzyme extract) was used to measure enzyme activity. SOD activity: The reaction mixture (reaction volume, 3 mL) contained 50 mM PBS (pH 7.8), 14.5 mM methionine, 2.25 mM NBT, 60 μM riboflavin, 30 μM EDTA, and 20 μL of enzyme extract. Subsequently, the mixture was allowed to react for 20 min under light (4000 lx) and the absorbance of the supernatant was measured at 515 nm. POD activity: A reaction mixture of 3 mL was prepared using 200 mM PBS (pH 6.0), 0.056% H_2_O_2_, 0.038% guaiacol, and 20 μL of enzyme extract. The absorbance of the supernatant was measured at 470 nm for 5 min. CAT activity: The reaction mixture (3 mL) contained 150 mM PBS (pH 7.0), 0.15% H_2_O_2_, and 15 μL of enzyme extract. Absorbance was measured at 240 nm for 5 min.

### 4.11. Statistical Analysis

SPSS 17.0 (IBM Corp., Armonk, NY, USA) software was used for data analysis, Duncan’s test was used to test significance (‘*’ represent *p* < 0.05, significant; ‘**’ represent *p* < 0.01, highly significant). All experiments were repeated thrice. All data are expressed as the mean ± one standard deviation (SD).

## 5. Conclusions

In this study, we identified six PAO genes in pepper, named *CaPAO1*–*CaPAO6* according to their chromosomal positions. PAO proteins were divided into four subfamilies by comparative phylogenetic analysis; CaPAOs were divided into subfamilies I, III, and IV, with none belonging to subfamily II. Based on their expression patterns, *CaPAO2*, *CaPAO4*, and *CaPAO6* may be involved in regulating pepper growth and development. The subcellular localization results showed that CaPAO2 and CaPAO4 proteins were localized in the peroxisome, and the results of substrate specificity showed that the preferred substrate was Spm. *CaPAO2* and *CaPAO4* overexpression in *Arabidopsis* enhances tolerance to freezing stress by altering the physiological and molecular levels in transgenic plants. This study provides a basis for exploring the regulation mechanism of cold reaction, but the specific mechanisms underlying the actions of *CaPAO2* and *CaPAO4* in pepper need to be studied further.

## Figures and Tables

**Figure 1 ijms-23-09999-f001:**
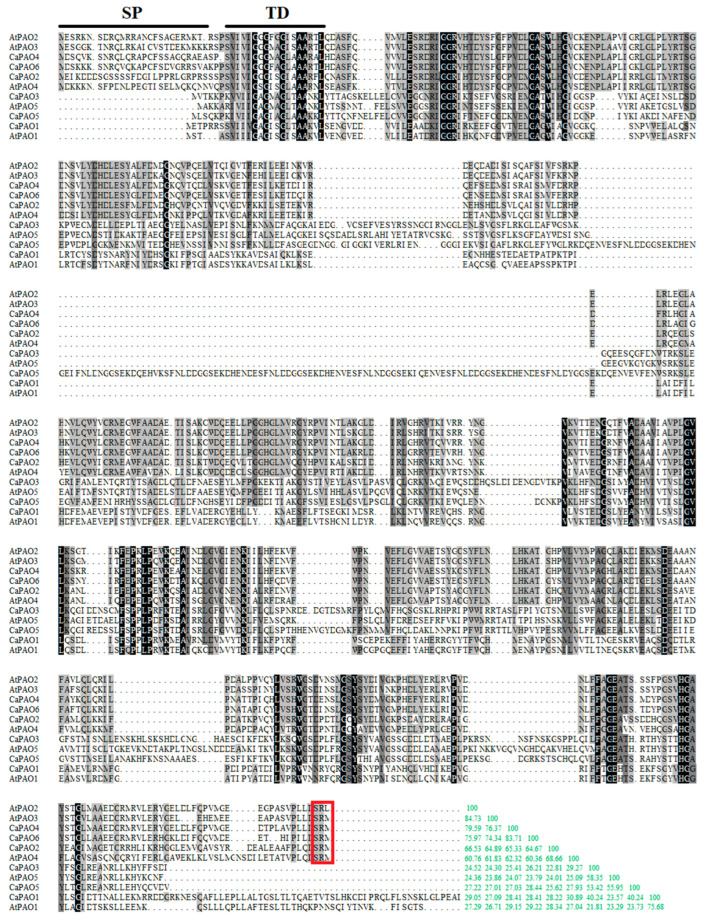
Alignment of pepper and *Arabidopsis* PAO protein sequences. Black lines indicate the signal peptide and transmembrane domain (SP + TD) [32]. Identical and similar residues are shaded in black and gray background, respectively. Peroxisomal targeting signals of PAOs are indicated in red box. The consistency of each amino acid sequence is shown at the end of the alignment, and the green number represents the percentage.

**Figure 2 ijms-23-09999-f002:**
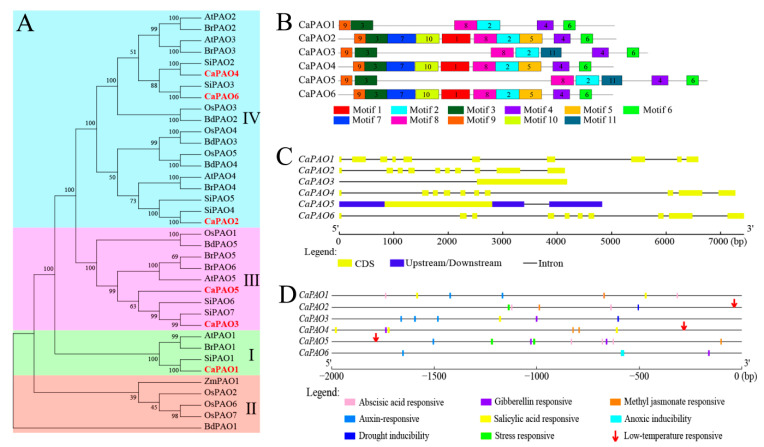
Multiple sequence alignment, motifs, gene structure, and cis-elements of CaPAO family members. (**A**) Phylogenetic tree of PAO protein domain sequences from pepper, *Arabidopsis*, tomato, rice, and other plants using the maximum likelihood method from the ClustalW alignment. At: *Arabidopsis* (*Arabidopsis thaliana*); Si: tomato (*Solanum lycopersicum* L.); Os: rice (*Oryza sativa*); Br: *Brassica rapa*; Bd: *Brachypodium distachyon*; Zm: maize (*Zea mays* L.). (**B**) Schematic distributions of conserved motifs among CaPAO genes. (**C**) Analysis of CaRboh gene structures. (**D**) Analysis of cis-elements in the CaPAOs genes promoter regions.

**Figure 3 ijms-23-09999-f003:**
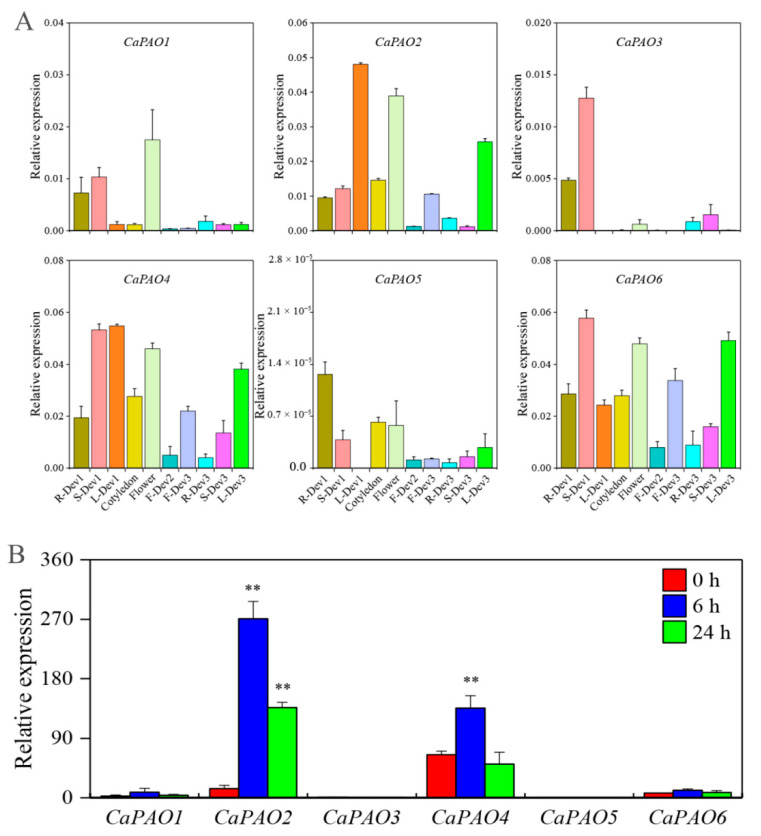
Expression analysis of *CaPAO* genes in various tissues and in response to cold stress. (**A**) qRT-PCR analysis of CaPAOs in different tissues. R-Dev1: root (six-leaf stage); S-Dev1: stem (six-leaf stage); L-Dev1: leaf (six-leaf stage); F-Dev2: fruit (Immature stage, green, fruits length between 1–2 cm); F-Dev3: fruit (mature stage, yellow, fruits length between 2–4 cm); R-Dev3: root (Fruit maturity); S-Dev3: stem (fruit maturity); L-Dev3: leaf (fruit maturity). (**B**) Transcriptome analysis of CaPAO genes in response to cold stress [30]. ‘**’ indicates a significant difference between the treatment and the control at *p* < 0.01.

**Figure 4 ijms-23-09999-f004:**
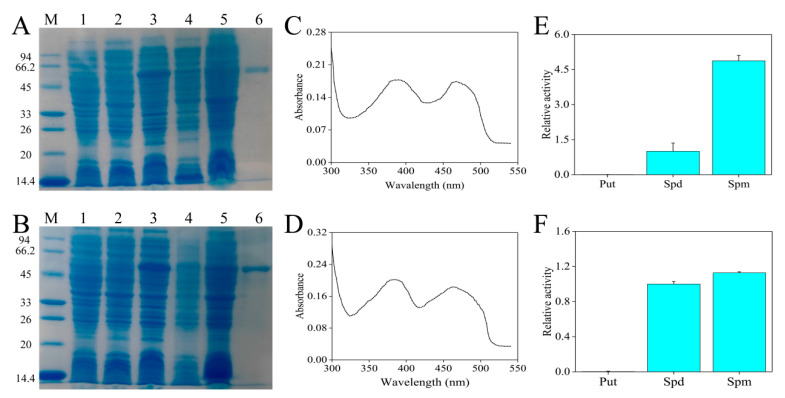
Heterologous expression of partial CaPAO proteins. SDS-PAGE analysis results of (**A**) CaPAO2 and (**B**) CaPAO4 protein prokaryotic expression. M: protein marker, 14.4–94 KDa; Lane 1: p-COLD; Lane 2: Induction without IPTG; Lane 3, crude lysate; Lane 4: Supernatant after pyrolysis; Lane 5: precipitation after pyrolysis; lane 6, purified recombinant protein. Absorption spectrum of the purified (**C**) CaPAO2 and (**D**) CaPAO4 in a range of 300–540 nm, respectively. PA specificity of recombinant (**E**) CaPAO2 and (**F**) CaPAO4. The PAO enzyme activity, with Spd as the substrate, was set to 1 to show the relative activities of other substrates.

**Figure 5 ijms-23-09999-f005:**
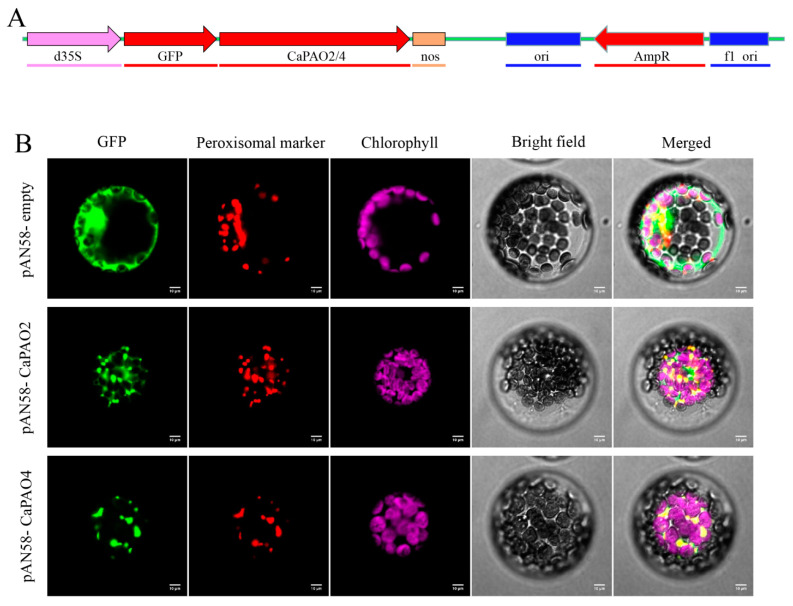
Subcellular localization of CaPAO2 and CaPAO4. (**A**) pAN58-GFP-CaPAO2/4 fusion vector construction. (**B**) Subcellular localization of CaPAO2 and CaPAO4 in *Arabidopsis* protoplasts. The localization of GFP and its fusion proteins is shown in green, the localization of peroxisomal marker is shown in red, and the chloroplast fluorescence is shown in purple. Scale bars = 10 mm.

**Figure 6 ijms-23-09999-f006:**
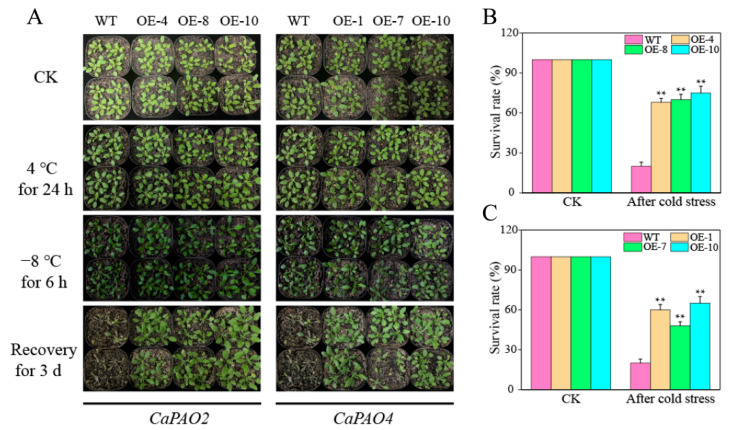
CaPAO2 and CaPAO4 overexpression in *Arabidopsis* enhanced freezing tolerance. (**A**) Phenotypes of wild-type (WT) and transgenic lines after low temperature stress. (**B**) Survival rates of WT and CaPAO2-OE lines after −8 °C for 6 h. (**C**) Survival rates WT and CaPAO4-OE lines after −8 °C for 6 h. “**” indicates *p* < 0.01 between transgenic lines and WT.

**Figure 7 ijms-23-09999-f007:**
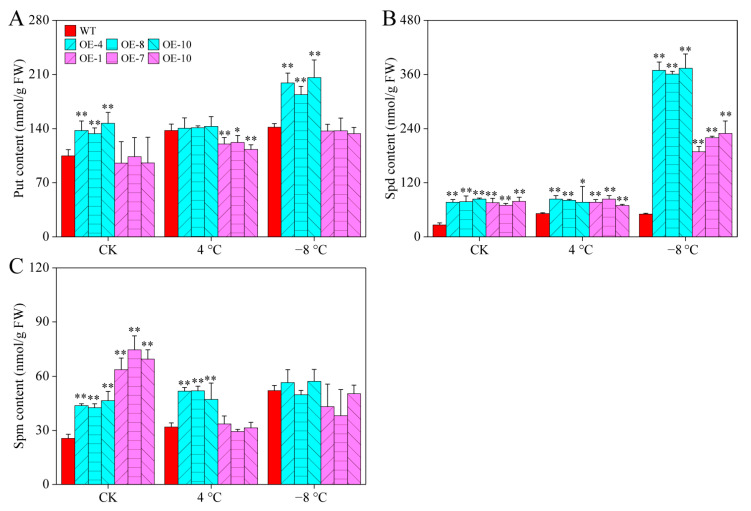
PA content in pepper under cold stress. (**A**) Put content. (**B**) Spd content. (**C**) Spm content. Red represents WT plants, blue represents *CaPAO2* transgenic plants, and purple represents *CaPAO4* transgenic plants. “*” indicates *p* < 0.05 between transgenic lines and WT; “**” indicates *p* < 0.01 between transgenic lines and WT.

**Figure 8 ijms-23-09999-f008:**
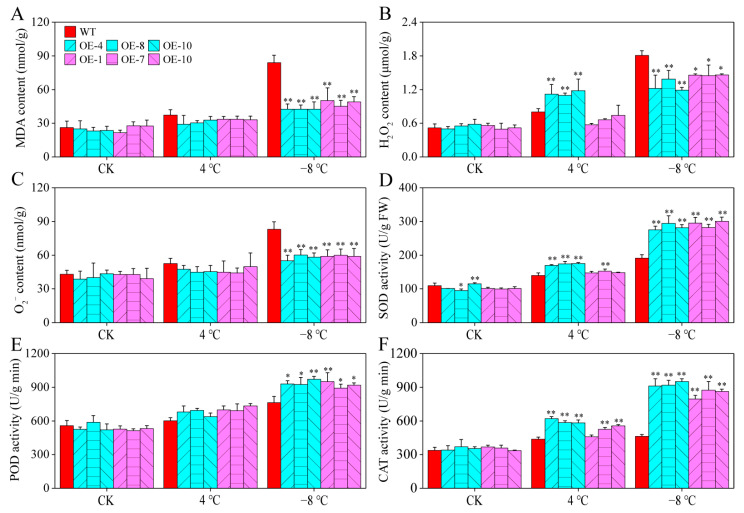
Determination of physiological parameters and antioxidant enzymes in pepper under cold stress. (**A**) MDA content. (**B**) H_2_O_2_ content. (**C**) O_2_^−^ content. (**D**) SOD activity. (**E**) POD activity. (**F**) CAT activity. Red represents WT *Arabidopsis* plants, blue represents *CaPAO2* transgenic plants, and purple represents *CaPAO4* transgenic plants. “*” indicates *p* < 0.05 between transgenic lines and WT; “**” indicates *p* < 0.01 between transgenic lines and WT.

**Figure 9 ijms-23-09999-f009:**
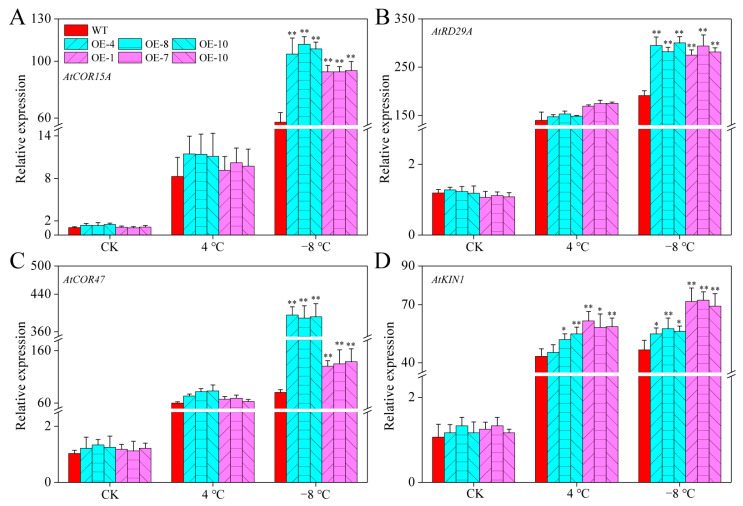
Expression levels of (**A**) AtCOR15A, (**B**) AtRD29A, (**C**) AtCOR47, and (**D**) AtKIN1 of transgenic lines and WT plants under cold stress. Atactin2 was used as the reference gene for *Arabidopsis*. Red represents WT *Arabidopsis* plants, blue represents *CaPAO2* transgenic plants, and purple represents *CaPAO4* transgenic plants. “*” indicates *p* < 0.05 between transgenic lines and WT; “**” indicates *p* < 0.01 between transgenic lines and WT.

**Table 1 ijms-23-09999-t001:** Details of the PAO gene family in pepper.

Gene Name	Gene Symbol	Protein ID	Protein Name	Protein Length (aa)	Protein Weight (kDa)	pI	Instability Index	Putative Location
*CaPAO1*	*LOC107858653*	NP_001311678.1	CaPAO1	496	55.8	5.22	38.64	Extracellular
*CaPAO2*	*LOC107859655*	XP_016560216.1	CaPAO2	495	54.8	5.65	37.78	Peroxisomal
*CaPAO3*	*LOC107870798*	XP_016572934.1	CaPAO3	551	70.0	5.43	35.46	Extracellular
*CaPAO4*	*LOC107878076*	XP_016580438.1	CaPAO4	490	54.7	5.91	36.91	Peroxisomal
*CaPAO5*	*LOC107840963*	XP_016540412.1	CaPAO5	657	72.5	4.74	36.06	Extracellular
*CaPAO6*	*LOC107841896*	XP_016541238.1	CaPAO6	489	54.5	5.74	36.35	Peroxisomal

## Data Availability

Not applicable.

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
