# Peer review of "Genome-Wide Identification of Polyamine Oxidase (PAO) Family Genes: Roles of CaPAO2 and CaPAO4 in the Cold Tolerance of Pepper (Capsicum annuum L.)"

_ijms, 2022, doi:10.3390/ijms23179999_

Round 1

Reviewer 1 Report

This is a very thorough study on an oxidase gene family in pepper and it is very well done. I only have a few comments on the methods section.

line 462: How are PFAM and CD used to remove redundant sequences exactly?

line 467: The exact parameters of the tools run in the analysis are not given. Therefore, any reader cannot reproduce the authors' results.

line 471: There are better tools for sequence alignment than ClustalX, such as MAFFT or MUSCLE. 

Author Response

Response to Reviewer 1 Comments

Point 1: How are PFAM and CD used to remove redundant sequences exactly?

Thank you very much for your comment. We removed the redundant sequence by whether it contained the amino_oxidase domain (PF01593) and whether the sequence was complete. We have added it in the materials and methods. The details were in the manuscript by marking in red.

Point 2: The exact parameters of the tools run in the analysis are not given. Therefore, any reader cannot reproduce the authors' results.

Thank you very much for your advice. We have added it in the materials and methods. The details were in the manuscript by marking in red.

Point 3: There are better tools for sequence alignment than ClustalX, such as MAFFT or MUSCLE. 

Thank you very much for your comment. First of all, I'm very sorry that we can't fully understand these three alignment tools. We used ClustalX tool for multi sequence alignment, it is mainly because of reading relevant literature [1-4] and referring to the operating methods of the laboratory. In addition, we also used the Clustal Omega and muscle tools to confirm the reliability of the results. Finally, thank you again for your comments.

[1] Lei, D.; Lin, Y.; Luo, M.; Zhao, B.; Tang, H.; Zhou, X.; Yao, W.; Zhang, Y.; Wang, Y.; Li, M.; et al. Genome-Wide Investigation of G6PDH Gene in Strawberry: Evolution and Expression Analysis during Development and Stress. Int. J. Mol. Sci. 2022, 23, 4728. https:// doi.org/10.3390/ijms23094728

[2] Zhang, Y.; Zhang, Y.; Luo, L.; Lu, C.; Kong, W.; Cheng, L.; Xu, X.; Liu, J. Genome Wide Identifification of Respiratory Burst Oxidase Homolog (Rboh) Genes in Citrus sinensis and Functional Analysis of CsRbohD in Cold Tolerance. Int. J. Mol. Sci. 2022, 23, 648. https://doi.org/10.3390/ ijms23020648

[3] Davoudi, M.; Chen, J.; Lou, Q. Genome-Wide Identification and Expression Analysis of Heat Shock Protein 70 (HSP70) Gene Family in Pumpkin (Cucurbita moschata) Rootstock under Drought Stress Suggested the Potential Role of these Chaperones in Stress Tolerance. Int. J. Mol. Sci. 2022, 23, 1918. https:// doi.org/10.3390/ijms23031918

[4] Wang, Z.; Ni, L.; Liu, D.; Fu, Z.; Hua, J.; Lu, Z.; Liu, L.; Yin, Y.; Li, H.; Gu, C. Genome-Wide Identifification and Characterization of NAC Family in Hibiscus hamabo Sieb. et Zucc. under Various Abiotic Stresses. Int. J. Mol. Sci. 2022, 23, 3055. https://doi.org/10.3390/ ijms23063055

Reviewer 2 Report

The manuscript by Zhang et al., performed an in silico identification of Polyamine oxidases gene family in pepper.  Manuscript is well structured. Need minor modifications before acceptance

The manuscript needs little modification as suggested

1.       Line 86-89, reference missing please cite relevant reference

2.       Line 106, subcellular > in silico subcellular

3.       For MSA why single gene from maize is used why not the preferred tomato belonging to the same group?Although the similarity level is too low to be negligible as stated in line 121.  I will suggest author should use either Arabidopsis or tomato PAO proteins for MSA.

Author Response

Response to Reviewer 2 Comments

Point 1: Line 86-89, reference missing please cite relevant reference

Thank you very much for your advice. We have added references in the manuscript, the details were marked in red.

Point 2: Line 106, subcellular > in silico subcellular

Thank you very much for your comment. We have revised the ‘subcellular’ to ‘in silico subcellular’. The details were in the manuscript by marking in red.

Point 3: For MSA why single gene from maize is used why not the preferred tomato belonging to the same group?Although the similarity level is too low to be negligible as stated in line 121.  I will suggest author should use either Arabidopsis or tomato PAO proteins for MSA.

Thank you for your good suggestions of our manuscript. Due to the relatively large number of PAOs in Arabidopsis and tomato, the PAO sequence of pepper can not be well reflected. In addtion, the cDNA and amino acid sequence of PAO in maize were first reported by Tavladoraki et al. [1] in 1998, and the primary structure of the mature protein was independently confirmed. Generally, maize PAO (ZmPAO) was used as the reference sequence for multiple sequence alignment of different species, such as Arabidopsis, tomato, sweet orange, and peach. Surely, the phylogenetic tree including tomato and Arabidopsis was constructed, and we found that pepper PAO proteins were more closely related to those from tomato and Arabidopsis than to those of maize and rice. Finally, thank you again for your comments.

[1] Tavladoraki P.; Schininà M.E.; Cecconi F.; Di Agostino S.; Manera F.; Rea G.; Mariottini P.; Federico R.; Angelini R. Maize polyamine oxidase: primary structure from protein and cDNA sequencing. FEBS Lett. 1998, 426, 62–66. https://doi: 10.1016/S0014-5793(98)00311-1

Round 2

Reviewer 1 Report

The authors have addressed my previous comments.

Author Response

Thank you very much for your comments on this article.

Reviewer 2 Report

I am not satisfied with the answer from point 3 which is about using maize as a reference instead of Arabidopsis and tomato. As I can see both Arabidopsis and tomato contained five PAO genes than pepper itself as shown in Table 1. I suggest authors draw MAS either using Arabidopsis or tomato protein sequence. additionally, figure out the reseason why pepper has 5 PAO 1 more than Arabidopsis and tomato. Is it a duplication of an existing gene and what is the biological significance of that additional one?

Author Response

Response to Reviewer 2 Comments

Point 3: I am not satisfied with the answer from point 3 which is about using maize as a reference instead of Arabidopsis and tomato. As I can see both Arabidopsis and tomato contained five PAO genes than pepper itself as shown in Table 1. I suggest authors draw MAS either using Arabidopsis or tomato protein sequence. additionally, figure out the reseason why pepper has 5 PAO 1 more than Arabidopsis and tomato. Is it a duplication of an existing gene and what is the biological significance of that additional one?

Thank you again for your good suggestions of our manuscript. It is really true as reviewer suggested that Arabidopsis or tomato is more suitable for MAS with pepper. We have revised the graph. The details were in the manuscript by marking in red. In this study, six PAO genes (CaPAO1-CaPAO6) were identified in pepper (named CaPAO1-CaPAO6 according to their chromosomal positions), which is similar to previous reports of Arabidopsis (5, AtPAO1-AtPAO5) [1] and tomato (7, SiPAO1-SiPAO7) [2]. Previously, no CaPAO homologous genes were identified in the pepper genome through Synteny analysis.

  1. Takahashi Y.; Cong R.; Sagor G.H.M.; Niitsu M.; Berberich T.; Kusano T. Characterization of five polyamine oxidase isoforms in Arabidopsis thaliana. Plant Cell Rep. 2010, 29, 955–965 [doi: 10.1007/s00299-010-0881-1].
  2. Hao Y.; Huang B.; Jia D.; Mann T.; Jiang X.; Qiu Y.; Niitsu M.; Berberich T.; Kusano T.; Liu T. Identification of seven polyamine oxidase genes in tomato (Solanum lycopersicum L.) and their expression profiles under physiological and various stress conditions. J. Plant Physiol. 2018, 228, 1–11 [doi: 1016/j.jplph.2018.05.004].